# Factors associated with unsafe abortion practices in Nepal: Pooled analysis of the 2011 and 2016 Nepal Demographic and Health Surveys

**Resham Bahadur Khatri**[1☯*¤], **Samikshya Poudel**[2☯], **Pramesh Raj Ghimire**[3☯]

**1** Center for Research and Development, Surkhet, Nepal, **2** Ujyalo Nepal, Ratnanagar Municipality, Chitwan, Nepal, **3** School of Science and Health, Western Sydney University, Sydney, Australia

☯ These authors contributed equally to this work.
¤ Current address: School of Public Health, Faculty of Medicine, University of Queensland, Brisbane, Australia
\* rkchettri@gmail.com

**Data Availability Statement:** Data used in this study are publicly available secondary data obtained from the DHS (https://dhsprogram.com/data/available-datasets.cfm) program.

## Abstract

### Background

Unsafe abortion contributes to maternal morbidities, mortalities as well as social and financial costs to women, families, and the health system. This study aimed to examine the factors associated with unsafe abortion practices in Nepal.

### Methods

Data were derived from the 2011 and 2016 Nepal Demographic and Health Surveys (NDHS). A total of 911 women aged 15–49 years who aborted five years prior to surveys were included in the analysis. The multivariate logistic regression analysis was employed to determine factors associated with unsafe abortion.

### Results

Unsafe abortion rate was seven per 1000 women aged 15–49 years. This research found that women living in the Mountains (adjusted Odds Ratio (aOR) 2.36; 95% CI 1.21, 4.60), or those who were urban residents (aOR 2.11; 95% CI 1.37, 3.24) were more likely to have unsafe abortion. The odds of unsafe abortion were higher amongst women of poor households (aOR 2.16; 95% CI 1.18, 3.94); Dalit women (aOR 1.89; 95% CI 1.02, 3.52), husband with no education background (aOR 2.12; 95%CI 1.06, 4.22), or women who reported agriculture occupation (aOR 1.82; 95% CI 1.16, 2.86) compared to their reference's group. Regardless of knowledge on legal conditions of abortion, the probability of having unsafe abortion was significantly higher (aOR 5.13; 95% CI 2.64, 9.98) amongst women who did not know the location of safe abortion sites. Finally, women who wanted to delay or space childbirth (aOR 2.71; 95% CI 1.39, 5.28) or those who reported unwanted birth (aOR = 2.33; 95% CI 1.19, 4.56) were at higher risk of unsafe abortion.

**Funding:** The author(s) received no specific funding for this work.

**Competing interests:** The authors have declared that no competing interests exist.

## Conclusion

Going forward, increasing the availability of safe abortion facilities and strengthening family planning services can help reduce unsafe abortion in Nepal. These programmatic efforts should be targeted to women of poor households, disadvantaged ethnicities, and those who reside in mountainous region.

## Introduction

World Health Organization (WHO) defines unsafe abortion as a procedure for terminating an unintended pregnancy, carried out either by persons lacking the necessary skills or in an environment that does not conform to minimal medical standards, or both [1]. Every year, approximately 25 million unsafe abortions occur worldwide; of these, 97% are reported in developing countries, and half of them in Asia [2]. Unsafe abortion plays an important role in maternal morbidity, disability and mortality; largely from post-abortion sepsis, haemorrhage, genital trauma, infection and infertility [3]. Recent estimates suggest that about 13% of global maternal deaths are attributed to unsafe abortion [4]. Also, approximately seven million women undergo treatment due to complications from unsafe abortion [5]; and about five million women suffer disability as a result of such complications [6]. Because of high maternal morbidity, mortality, and disability caused by unsafe induced abortion, the 57th World Health Assembly endorsed unsafe abortion as a major public health concern in 2004; since then, eliminating unsafe abortion has become an important agenda for WHO global strategy on reproductive health [7]. This global strategy also suggested that eliminating unsafe abortion would require scientific evidence to formulate relevant policies and programs.

Globally, the underlying causes of unsafe abortion are unmet need for family planning and unintended pregnancy [8]. In developing countries, women often choose unsafe abortion services to end unintended pregnancies [3]. Unsafe abortion rate is estimated to be 16 per 1000 women in low- and middle- income countries, which is slightly lower than South-Central Asian region (estimated to be 17 per 1000 )[3]. Unsafe abortion rate and related complication are high when countries lack legal access to abortion and/or have no institutional provision for safe abortion services [9]. Studies conducted in LMICs of African and Latin American region reported that unsafe abortion rate was higher among women with lower income, ethnic minorities, and lower education [10–12].

In South Asia, Nepal has become a pioneer in legalization, implementation and scale-up of safe abortion services [13]. In 2002, the Nepalese government granted women the right to abortion up to a specific gestational age-dependent upon circumstances or medical conditions. For instance, women can terminate pregnancy on request within the first 12 weeks of gestation. In case of rape or incest, pregnancy can be terminated up to 18 weeks of gestation. If a doctor recommends that the pregnancy poses a danger to the life, physical or mental health of the pregnant woman, or if the fetus is seriously deformed, then abortion can be done any time of gestation [14]. Following this legal reform, a comprehensive safe abortion care program was implemented in 2004 [13]. In 2009, after the feasibility study of safe induced medical abortions services for pregnancies up to 9 weeks of gestational age, the phase-wise scaling up of the program was initiated in rural health posts with birthing centres facilities by skilled birth attendants (auxiliary nurse midwives having two months training on safe childbirth skills)[15, 16]. Until 2017, medical abortion services were available in 49 districts (out of 77 districts) [17, 18].

Abortion services are provided in certified health facilities by doctors and skilled birth attendants trained on abortion services [18–20].

After more than decade-long programmatic responses, the utilization of safe abortion services has not yet been universally adopted in Nepal. For instance, in 2014, out of total estimated 323,000 abortions, about 58% of abortions were conducted using a clandestine procedure provided by untrained/uncertified health providers or induced by the pregnant woman herself [19]. Previous literature has documented that there are challenges for the delivery of abortion services that include limited coverage of abortion sites, lack of trained human resources, and necessary equipment and medicines in accredited health facilities [19]. A lower contraceptive prevalence rate (53%) and higher unmet need for family planning services (24%) [21] resulted in high unintended pregnancy[21] that could potentially compel women to use unsafe abortion services. Also, a qualitative study in Nepal reported that abortion service seekers experienced denial from safe abortion services due to higher gestational age, and these women adopted unsafe abortion practices [22]. Women who sought abortion services had lower knowledge on the location of certified abortion sites[23] as well as legal conditions of abortion with higher unintended and untimed pregnancies [24–26].

Additionally, women who reported unsafe abortion were less likely to know the legal provision of abortion in Nepal compared with those who reported safe abortion services [25]. A recent study conducted in Nepal revealed that women of higher socioeconomic status had lower odds of unsafe abortion practices [27]. However, this study is insufficient to unpack the contributing factors for the needs of unsafe abortion practices, including knowledge on safe places for abortion services. There is a dearth of knowledge gaps in the role of enabling and modifiable factors that could be useful to revise the abortion policies and practices in Nepal.

This suggests the scientific evidence is needed to revisit the existing policies and programs for eliminating unsafe induced abortions practices in Nepal. The WHO suggests that empirical research on unsafe abortion would help to re-evaluate existing programs as well as formulate appropriate strategies to improve safe abortion services[1, 3, 28]. Hence, this study aimed to provide a national estimate on the unsafe abortion rate and examine factors associated with unsafe abortion using the data from the Nepal Demographic and Health Surveys (NDHS) 2011 [29] and 2016 [21]. The findings from this study would open up discussion around evaluating existing abortion policies and programs and designing targeted strategies to eliminate unsafe abortion and achieve the maternal health-related target of 3.2 of Sustainable Development Goals (SDGs) 3[30].

## Methods

### Data sources

This research has derived data from the NDHS 2011 and 2016 (available from https://dhsprogram.com/data/new-user-registration.cfm). NDHS is also part of the Demographic and Health Survey Program. The DHS program is US Government-funded global health program, provides technical and financial support to conduct demographic and health surveys and health facility surveys in more than 90 LMICs around the globe. These surveys are implemented in partnership with ICF International (USA based company) and the government of the host country. In Nepal, under the leadership of the Ministry of Health and Population and technical support from ICF international, New Era (local research organization) conducts the NDHS in every five years[21, 29, 31].

## Sample

Data used in this analysis were based on women's questionnaires. The NDHS used two-stage cluster random sampling. A total of thirteen strata were constructed using five development regions and three ecological regions. In the first stage, the primary sampling units, wards of rural and sub-wards of urban areas of each stratum were selected, which also called as Enumeration Areas (EAs). In the second stage, households were selected using simple random sampling technique. The details of sampling techniques are further described in the full report of NDHS 2011 and 2016.

The data of NDHS 2011 and 2016 were merged to get the maximum sample size for this study. A total of 25, 536 women of reproductive age (15–49 years) were interviewed in the two surveys (NDHS 2011 and 2016). The average response rates for women aged 15–49 years in the NDHS 2011 and 2016 were 98%. Women who received the most recent abortion services five years prior to the surveys constituted study population. A total of 911 women received abortion services during the survey period.

## Outcome variable

In the surveys, information on the abortions services was collected using the following questions. In the pregnancy history section of the questionnaire, women were asked: Did you, or someone else do something to end this pregnancy?' has a yes/no response. Women responding 'yes' are then asked further questions about their abortion.

The outcome variable for this study was 'unsafe abortion'. Based on WHO definition [1], unsafe abortion was coded as '1' if the pregnancy was terminated either by persons lacking the necessary skills or in an environment that does not conform to minimal medical standards or both; otherwise coded as "0". To comply with this definition, Nepal's abortion law [13], and previous literature [27], this research considered unsafe abortions if conducted by other than physicians and nurse-midwives or those carried out outside health facilities.

## Independent variables

Past studies conducted in Ghana [10], Ethiopia [12], Mexico [11] on factors associated with unsafe abortion, and the information available in datasets were employed as a basis for the selection of potential confounding variables [Fig 1]. Some variables such as ethnicity, wealth status, and knowledge of safe abortion place or legal conditions of abortion were further categorized for this study. For instance, the Government of Nepal has categorized ethnicities into six broad groups [32]: i) Dalit (Hill and Terai)); ii) Janajati (Indigenous Hill and Terai); iii) Madhesi (non-Dalit Terai caste groups); iv) religious minorities (Muslims); v) upper caste groups (Brahman/Chhetri) vi) Others (Thakuri and Sanayshi). Based on similar socioeconomic and geographical similarities, and other literature [33, 34] ethnicities were categorized into four groups: a) Brahman/Chhetri (merging with "Others" category); b) Dalit; c) Janajati; d) Madhesi (merging with "Muslims" category). Similarly, knowledge about certified abortion sites and legal conditions of abortion were categorised as: i) knew the legal condition of abortion and place for safe abortion, ii) only know the legal conditions, iii) only knows the location of the place for safe abortion, and iv) did not know both. In NDHS, wealth quintiles were calculated using principal component analysis of more than 40 households' asset items. In this research, households' wealth quintile were categorised into three groups: the bottom 40% was referred to as poor households, the next 40% as the middle households and the top 20% as rich households, consistent with previous studies [35, 36].

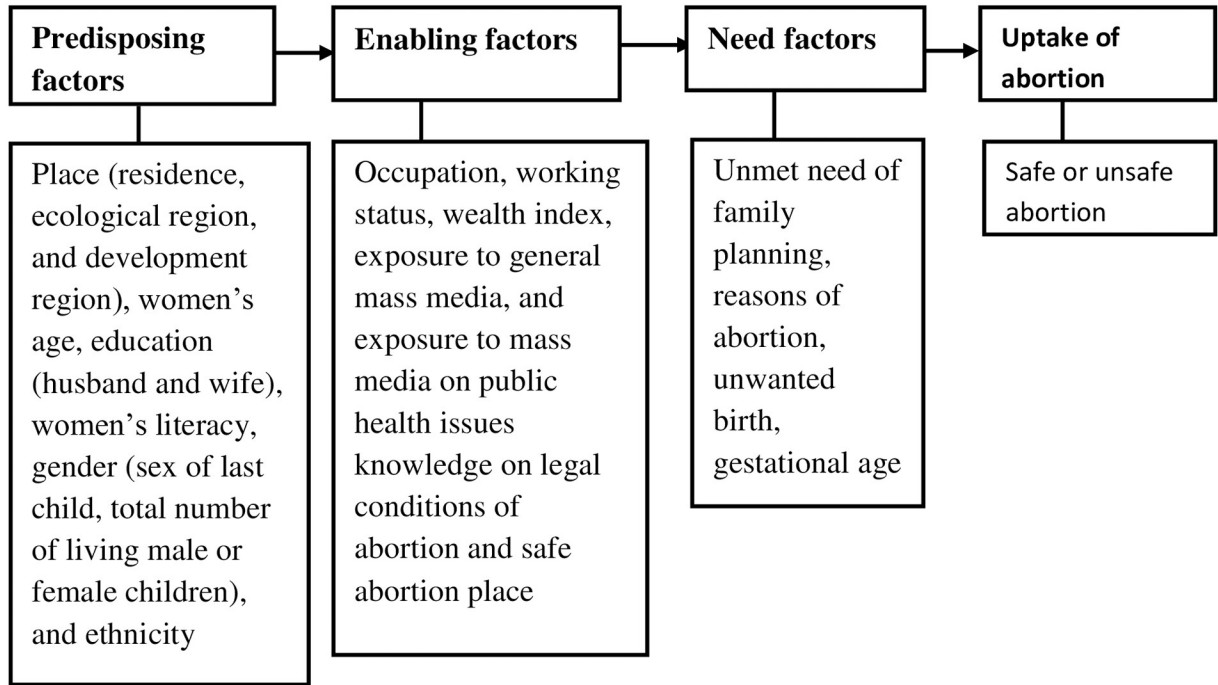

**Fig 1. The conceptual framework of factors of unsafe abortion adapted from modified Anderson's behavioural model** [37].

## Conceptual framework

A modified Anderson's behavioural model of health service use [37], which has been consistently used in other studies [38, 39], was adopted as a conceptual framework for this analysis [Fig 1]. According to this model, predisposing, enabling and need factors contribute to use/non-use of any health services.

Fig 1 shows the predisposing, enabling, and need factors of unsafe abortion services. *Predisposing factors* are existing conditions (not directly responsible for use) that predispose women to use or not abortion services. In this study, place of residence, women's age, the socioeconomic status including women education (women and their husbands), literacy status, ethnicity, gender (sex of the last-child), the total number of living son or daughters were considered as predisposing factors. Similarly, *enabling factors* are conditions that facilitate or impede the use of services. In this research enabling factors for unsafe abortion were household wealth index, occupation, mass media exposure, knowledge of legal conditions and certified abortion sites. *Need factors* are needs or conditions that women compel to use the services. In this study, the unmet need for family planning or unintended pregnancy, women's reasons for abortions, and gestational age at abortion were considered as need factors [Fig 1].

## Statistical analysis

Statistical analysis employed descriptive and staged regression models. Firstly, descriptive statistics such as frequencies and proportions were calculated to provide population-based estimates of the outcome variable. Abortion rates were calculated considering the definition of total number of abortion (safe or unsafe abortion) occurring in a specified period per 1,000 women aged15-49 years [3]. This research estimated the rates of abortion and unsafe abortion and their 95% Confidence Interval (CI). Secondly, staged logistic regression [40–42] models were conducted to examine factors associated with unsafe abortion while adjusting for

potential confounding factors. Unadjusted odds ratios and their 95% CI were calculated to examine the association between each independent variable and unsafe abortion (model 1).

Before moving to the multivariate logistic regression analysis, multi-collinearity was checked using variation inflation factors (VIF) test considering VIF cut-off value >3[43] (none of the independent variables was found cut-off values> 3). At the second stage, the predisposing factor was entered and used manual backward elimination technique to retain statistically significant variables associated with unsafe abortion at 5% significance level (model 2). The same procedure was followed when enabling, and need factors added in the third stage (model 3), and the final stage (model 4), respectively. Factors significantly associated (p<0.05) with unsafe abortion in the final model (model 4) was reported [34]. To confirm/validate the result of the staged regression model, other alternative logistic regressions were also conducted [34, 36] a) entering only potential risk factors with p-value < 0.20 obtained in the bivariate analysis for backward elimination process, and b) testing the backward elimination method by including all potential risk factors. Complex sample analyses technique was adopted throughout to account for the study design, and sample weight, and analysis [36, 44]. A total of 45 missing values were excluded from the multivariate analysis. All analyses were performed in STATA (Stata Corp, College Station, Texas US) software version 14.0.

### Ethics approval

These surveys were approved by an ethical review board of Nepal Health Research Council, Nepal, and ICF Marco International, Maryland, USA. The first author got permission from DHS program (USA) to use those datasets for this study.

## Results

### Descriptive characteristics of the study population

Out of 911 women who used abortion services during 2011–2016, slightly over 50% were living in rural areas [Table 1]. Overall, 50% of the women and 72% of their husbands had secondary and higher level of education. Having access to general mass media and knowledge of safe abortion place were almost equally distributed (91% and 90% respectively).

### Abortion practices

Out of 25,536 women surveyed during the period (2011–2016), 911 women used abortion services; and of these abortion services, 23% (236) were unsafe abortions. The rate of abortion was estimated as 36 (95% CI: 33, 38) per 1000 women aged 15–49; whereas the rate of unsafe abortion was seven (95% CI: 6, 8) per 1000 women aged 15–49 years [Table is not shown].

### Descriptive analysis of unsafe abortion

The majority (17%) of the abortions were below eight weeks of gestational age(Table 1). A substantial proportion of unsafe abortions were conducted in the Mountain region (39%), and among those with the disadvantaged ethnic background (Dalit, and Madheshi and Muslim). Similarly, a higher proportion of women were found to undertake unsafe abortion practices if they or their husbands reported no education (36%), if they could not read or write, belonged to the households of lower wealth index, or were involved in agricultural occupation. If women had lower knowledge of legal conditions and safe abortion places (62%), or if they had no exposure to mass media, then a higher proportion of women used unsafe abortion services. If women wanted to delay or space childbirth or did not want birth, then a higher proportion of women were found to use unsafe abortion services [Table 1].

**Table 1. Descriptive characteristics of the study population and the proportion of unsafe abortion in Nepal, 2011–2016 (N = 911).**

| Variables | Categories | Total abortion | Unsafe abortion (%) | P |
|---|---|---|---|---|
| Total population | | 911 | 236 (26) | |
| **Predisposing factors** | | | | |
| Rurality | Rural | 495 | 107(22) | 0.008 |
| | Urban | 416 | 129(31) | |
| Eco-region | Hill | 419 | 93(22) | 0.035 |
| | Terai | 438 | 122(28) | |
| | Mountain | 54 | 21(39) | |
| Development region | Western | 268 | 62(23) | 0.193 |
| | Central | 240 | 58(24) | |
| | Eastern | 153 | 38(25) | |
| | Mid-western | 136 | 48(36) | |
| | Far-western | 115 | 30(26) | |
| Women's age | 34–49 years | 216 | 51(24) | 0.664 |
| | 20–34 years | 652 | 174(27) | |
| | <20 years | 43 | 11(24) | |
| Ethnicity | Brahmin/Chettri | 402 | 84(21) | <0.001 |
| | Dalit | 119 | 45(38) | |
| | Janajati | 281 | 65(23) | |
| | Madhesi and Muslim | 109 | 42(38) | |
| Women's education level | Secondary or higher | 459 | 97(21) | 0.012 |
| | Primary | 216 | 65(30) | |
| | No education | 236 | 74(31) | |
| Women's literacy level | Can read part or whole of the sentence | 719 | 169(23) | 0.004 |
| | Cannot read | 192 | 67(35) | |
| Numbers of male children | None | 216 | 41(19) | 0.054 |
| | One | 417 | 112(27) | |
| | Two or more | 278 | 84(30) | |
| Numbers of female children | None | 290 | 87(30) | 0.035 |
| | One | 337 | 69(21) | |
| | Two or more | 284 | 79(28) | |
| Sex of the most recent child | Male | 509 | 149(29) | 0.043 |
| | Female | 361 | 80(22) | |
| Husband education | Secondary or higher | 659 | 148(23) | <0.001 |
| | Primary | 168 | 59(35) | |
| | No education | 76 | 28(36) | |
| **Enabling factors** | | | | |
| Wealth index | Rich | 206 | 33(16) | 0.002 |
| | Middle | 376 | 95(25) | |
| | Poor | 329 | 108(33) | |
| Women's occupation | Skilled | 262 | 46(18) | 0.007 |
| | Agriculture | 418 | 124(30) | |
| | Not working | 231 | 66(28) | |
| Women's working status | Currently working | 579 | 146(25) | 0.591 |
| | Currently not working | 332 | 90(27) | |
| Exposure to general mass media | No | 80 | 36(44) | <0.001 |
| | Yes | 831 | 200(24) | |
| Exposure to mass media on public health issues | No | 174 | 66(38) | <0.001 |

*(Continued)*

**Table 1.** (Continued)

| Variables | Categories | Total abortion | Unsafe abortion (%) | P |
|---|---|---|---|---|
| | Yes | 737 | 170(23) | |
| **Need factors** | | | | |
| Unmet need for family planning | No unmet need | 602 | 155(26) | 0.861 |
| | Unmet need | 309 | 81(27) | |
| Knowledge of condition and place of safe abortion | Knows condition and place for safe abortion | 610 | 131(21) | <0.001 |
| | Knows condition only | 57 | 36(63) | |
| | Knows place only | 212 | 49(23) | |
| | Absence of both | 32 | 20(62) | |
| Reason for abortion | Health of women | 94 | 14(15) | <0.001 |
| | Wanted to delay/spacing | 174 | 57(32) | |
| | Unwanted birth | 403 | 127(32) | |
| | Low family earning and others^£ | 240 | 38(16) | |
| Gestation(N = 735) | Up to 8 weeks | 580 | 150(26) | 0.583 |
| | 9–12 weeks | 117 | 26(22) | |
| | 13 weeks and more | 38 | 8(21) | |

P-value obtained from Chi-square association

## Factors associated with unsafe abortion practices in Nepal

Table 2 shows the results of bivariate and multivariate regression analyses of independent variables and unsafe abortion in Nepal. The bivariate logistic regression showed that rurality (urban), eco-region (Mountain), development region (mid-western), wealth index (middle or poor), ethnicity (Dalit, or Madhesi and Muslim), maternal education (primary or no education), women's literacy level (cannot read), husband education (primary or no education), maternal occupation (agriculture or no occupation), knowledge on legal conditions of abortion and safe abortion sites, exposure to general mass media (yes), and exposure to mass media on public health issues, number of male children ($\geq 2$), number of female children (one), sex of the most recent children (female), reasons for abortion (want to delay/space child-bearing, or unwanted child) were all significantly associated with unsafe abortion at $p<0.05$ [Table 2].

The final regression model [Table 2] revealed that women residing in the mountain region (aOR 2.36 95% CI 1.21, 4.60), or rural women (aOR 2.11, 95% CI 1.37, 3.24) were predisposed to unsafe abortion compared their hill or urban peers [Table 2].

Enabling factors such as women belonging to poor household had higher odds of having unsafe abortion (aOR 2.16, 95% CI 1.18, 3.94) compared to women of wealthy households. Additionally, unsafe abortion were significantly higher among Dalit (aOR 1.96, 95% CI 1.08, 3.54), Madhesi or Muslims (aOR 1.71, 95% CI 1.01, 2.88) compared to Brahmin/ Chhetri ethnic group. Husbands with no education (aOR 2.12 95% CI 1.06, 4.22), and women having occupation in agricultural sector (aOR 1.82 95% CI 1.16, 2.86) had higher odds of unsafe abortion compared to husband with secondary and higher education and women with skilled occupation respectively [Table 2].

Need factors such as knowledge on safe abortion places and legal conditions, and reasons for abortions were also significantly associated with unsafe abortion practices in Nepal. Women who did not know the place for safe abortion services (aOR 5.13 95% CI 2.64, 9.98) (but know legal conditions of abortions), and who did not know both (legal conditions of abortions and place for safe abortion) had higher odds of unsafe abortion practices compared

**Table 2. Unadjusted and adjusted odds ratio of factors associated with unsafe abortion in Nepal in 2011–2016 (N = 911).**

| Variables | Categories | Unadjusted OR (95% CI) | P | Adjusted OR (95% CI) | P |
|---|---|---|---|---|---|
| **Predisposing factors** | | | | | |
| Rurality | Rural | 1.00 | | 1.00 | |
| | Urban | 1.63(1.13, 2.36) | 0.009 | 2.11 (1.37, 3.24) | <**0.03** |
| Eco-region | Hill | 1.00 | | 1.00 | |
| | Terai | 1.35(0.91, 2.00) | 0.140 | 1.47(0.98, 2.21) | **0.063** |
| | Mountain | 2.22(1.27, 3.88) | 0.005 | 2.36(1.21, 4.60) | **0.012** |
| Development region | Western | 1.00 | | | |
| | Central | 1.04(0.60, 1.81) | 0.890 | | |
| | Eastern | 1.10(0.61, 1.97) | 0.747 | | |
| | Mid-western | 1.84(1.11, 3.02) | 0.017 | | |
| | Far-western | 1.51(0.64, 2.07) | 0.637 | | |
| **Predisposing factors** | | | | | |
| Women's age | 34–49 years | 1.00 | | | |
| | 20–34 years | 1.18(0.80, 1.75) | 0.399 | | |
| | <20 years | 1.04(0.48, 2.27) | 0.914 | | |
| Ethnicity | Brahmin/Chettri | 1.00 | | 1.00 | |
| | Dalit | 2.32(1.32, 4.07) | 0.004 | 1.89 (1.02, 3.52) | **0.043** |
| | Janajati | 1.13(0.76, 1.70) | 0.535 | 1.35 (0.90, 2.03) | 0.146 |
| | Madhesi and Muslim | 2.37(1.45, 3.86) | 0.001 | 2.10 (1.25, 3.54) | **0.005** |
| Women's education level | Secondary or higher | 1.00 | | | |
| | Primary | 1.60(1.05, 2.43) | 0.028 | | |
| | No education | 1.71(1.15, 2.57) | 0.009 | | |
| Women's literacy level | Can read part or whole of the sentence | 1.00 | | | |
| | Cannot read | 1.74(1.19, 2.54) | 0.004 | | |
| Husband education | Secondary or higher | 1.00 | | 1.00 | |
| | Primary | 1.87(1.20, 2.91) | 0.006 | 1.72(1.07, 2.75) | **0.024** |
| | No education | 1.98(1.12, 3.48) | 0.018 | 2.12(1.06, 4.22) | **0.033** |
| Numbers of male children | None | 1.00 | | | |
| | One | 1.58(0.95, 2.61) | 0.076 | | |
| | Two or more | 1.75(1.08, 2.83) | 0.023 | | |
| Numbers of female children | None | 1.00 | | | |
| | One | 0.63(0.41, 0.98) | 0.040 | | |
| | Two or more | 0.88(0.58, 1.33) | 0.536 | | |
| Sex of the most recent child | Male | 1.00 | | | |
| | Female | 0.70(0.49, 0.99) | 0.042 | | |
| **Enabling factors** | | | | | |
| Wealth index | Rich | 1.00 | | | |
| | Middle | 1.75(1.00, 3.03) | 0.047 | 1.70(0.91, 2.87) | 0.112 |
| | Poor | 2.52(1.50, 4.24) | 0.001 | 2.16 (1.18, 3.94) | **0.043** |
| Women's occupation | Skilled | 1.00 | | 1.00 | |
| | Agriculture | 1.94(1.25, 3.01) | 0.003 | 1.82(1.16, 2.86) | **0.009** |
| | Non- agriculture | 1.84(1.18, 2.88) | 0.008 | 1.53(0.93, 2.50) | **0.092** |
| Women's working status | Currently working | 1.00 | | | |
| | Currently not working | 1.09(0.79, 1.52) | 0.592 | | |
| Exposure to general mass media | No | 1.00 | | | |
| | Yes | 0.40(0.24, 0.66) | <0.001 | | |
| Exposure to mass media on public health issues | No | 1.00 | | | |

*(Continued)*

**Table 2.** (Continued)

| Variables | Categories | Unadjusted OR (95% CI) | P | Adjusted OR (95% CI) | P |
|---|---|---|---|---|---|
| | Yes | 0.49(0.33, 0.71) | <0.001 | | |
| **Need factors** | | | | | |
| Unmet need for family planning | No unmet need | 1.00 | | | |
| | Unmet need | 1.03(0.71, 1.50) | 0.862 | | |
| Knowledge of condition and place of safe abortion | Knows condition and place for safe abortion | 1.00 | | 1.00 | |
| | Knows legal conditions but not place | 6.34(3.41, 11.77) | <0.001 | 5.13(2.64, 9.98) | **<0.001** |
| | Knows place but not legal conditions | 1.10(0.73, 1.65) | 0.652 | 1.34(0.88, 2.03) | 0.172 |
| | Absence of both | 6.00 (2.81, 12.81) | <0.001 | 4.83(2.20, 10.61) | **<0.001** |
| Reason for abortion | Health of women | 1.00 | | 1.00 | |
| | Wanted to delay/spacing | 2.75(1.43, 5.32) | 0.003 | 2.71(1.39, 5.28) | **0.003** |
| | Unwanted birth | 2.66(1.36, 5.19) | 0.004 | 2.33(1.19, 4.56) | **0.014** |
| | Low family earning and others[£] | 1.08(0.53, 2.19) | 0.831 | 1.36(0.64, 2.89) | 0.418 |

Bold values indicate significance in the final model at p<0.05.

[£] Others category also include a reason such as no one in the family to look after the child, and to avoid shame.

with those who did know both. Finally, women who had unwanted pregnancy or wanted to delay or space childbirth had higher odds of unsafe abortion practices [Table 2].

## Discussion

This study revealed that the rates of abortion and unsafe abortion over the study period (2011–2016) were 36 and seven per 1000 women aged 15–49 years respectively. Independent variables such as eco-region, rurality, ethnicity, wealth index, husband education or women's occupation, knowledge on legal conditions of abortions and place for safe abortion, reasons of abortion were significantly associated with unsafe abortion.

The higher risk of unsafe abortion in the Mountain region may be aggravated due to difficult geographic terrain that may hamper both the access and utilization of safe abortion services. Availability of abortion services is limited to district hospitals or primary health care centres in the mountainous districts. Though medical abortion services have been available up to the health post level (birthing centre- health post having childbirth facilities only), many mountainous districts have not been covered by medical abortion services [32]. Women have to spend several hours to reach health facilities to get safe abortion services [20]. In addition, even health facilities are certified as abortion sites, unavailability of trained human resources, equipment, drugs are other challenges that bar safe abortion services in the Mountainous region could be the challenge [20]. In agreement with previous studies conducted in Nepal [27] and Tanzania [44], this study found that women living in rural Nepal were at higher risk of unsafe abortion.

Compared to other ethnic and religious groups, abortion practices are religiously stigmatized in Muslim communities, and culturally taboo in Madhesi and Dalits[19, 45]; and post-abortion women are often labelled as sinners (Papini), ill-luck (alichhini), murderers (jyanmaara), and foetus killers (garbhaghati) [19]. The higher odds of unsafe abortion amongst Muslim women in this study may be due to these cultural barriers that make women use abortion services other than certified health facilities or trained providers. In Nepal, the contraceptive prevalence rate is low; whereas, the unmet need for family planning is high [46]. The lower contraceptive prevalence rate and the higher unmet need for family planning are

considered as important contributors to unwanted pregnancy-a possible reason for unsafe abortion as documented in public health literature [8]. In Nepal, people from Dalit ethnic background and those who live in the Terai are relatively poor that makes access to safe abortion services further hard as the provision of free abortion services is not yet universal in Nepal [20].

This study identified significant differences in unsafe abortion practices based on different socioeconomic status. For instance, women having occupation in agricultural sectors, husbands with no education background, and women belonging to the households of lower wealth quintile were all significantly associated with unsafe abortion. These findings were similar to the studies conducted in Brazil [47] and Mexico [11], which also found that unsafe abortion was higher among women of lower-income, and women with low-level education. In Mexico, the legal status of abortion varied by state; Mexico city offers abortion up to 13 weeks gestation, whereas in Brazil abortion is legal if pregnancies result from rape or incest or if the life of the pregnant woman or fetus is at risk [48]. Both studies argued that the legal barriers to safe abortion services meant poor women could not afford quality abortion services, and they were compelled to use unsafe induced abortion. However, in Nepalese context, higher unsafe abortion practice among women of lower wealth status might be the financial inaccessibility to the safe abortion services as it was only made free of cost after 2017 [20]. Women from poor households were not able to get safe abortion services as women were required to pay at least 800–1200 Nepalese Rupees (8–12 USD) as service charge excluding medications (until data collection for NDHS 2016) [20]. In addition to the direct cost of abortion services, women are also required to pay other indirect costs such as cost for medicine, transportation, meal and accommodation [19, 20].

In contrast to conditions of Brazil [47] and Mexico [11], Nepal has overcome the legal barrier, but higher unsafe abortion is more prevalent among poor women. Higher unsafe abortion among poor socioeconomic groups in this study may be due to the need for family planning services. Socioeconomically disadvantaged and ethnic minorities groups in Nepal have lower contraceptive prevalence rates and higher unmet need for family planning services [21, 49]. Poor access and utilization of family planning services may lead to the use of abortion services as methods of spacing or delaying childbirth. However, women may not know the authorized place and legal conditions for abortion services [26], which possibly lead to unsafe abortion services.

The current study identified that women who did not know the place of safe abortion, regardless of their knowledge on legal conditions to have an abortion, had a higher likelihood of unsafe abortion practices. Previous studies conducted in Nepal revealed that women who were not aware of the legal provision (such as aborting period) or location of nearest safe abortion sites [23, 26] were more likely to have unsafe induced abortion. These facts show that being aware of certified abortion sites is important for the uptake of safe abortion services in Nepal.

In this study, though unmet need for family planning services was not significantly associated with unsafe abortion, the higher odds of unsafe abortion practices were significantly associated with child spacing or unwanted pregnancy. This indicates the need for family planning services to prevent unintended pregnancy. In Nepal, 24% of women had an unmet need for family planning (16% want to delay, and 8% want to space the birth), and 19% childbirth is from unwanted [21]. Evidence from Ghana suggests that unsafe abortion were higher if women have an unintended pregnancy [8]. Therefore, strengthening family planning services and reducing unintended pregnancy could be one of the strategies for reducing unsafe abortion in Nepal.

This study has some strengths and limitations. We pooled the data from nationally representative surveys conducted in the past decade. Thus, estimates from this study are generalizable

to the Nepalese population and can inform national policies and practices. Secondly, the response to the surveys was high (>98%), reducing a likely chance of selection bias from the observed findings. However, there might be recall bias because the information was collected through the recall of past experiences, and the recall period was long (5 years) that many increase the potential for misclassification of cases. Due to the small sample size, this study could not do a separate analysis for each of the survey wave (NDHS 2011 and NDHS 2016) for absolute comparison. It is an analysis of quantitative data and lacks qualitative information to explain the behaviour of women. Hence, future qualitative studies are needed to explore more inclusive intervention for culturally diverse population across the country.

## Policy and program implications

This study has policy and program implications. The legalization of abortion was the first move, but that does not seem sufficient enough for the delivery and utilization of safe abortion services[50]. Therefore, the increase in certified safe abortion sites and the provision of safe abortion services for women of the Mountainous region and socioeconomically disadvantaged groups could be an appropriate step to reduce unsafe abortion practices. From the demand side perspective, the community needs to be informed and sensitised about the use of safe abortion services[51]. Moreover, the integration of awareness-raising interventions in existing health programs could increase the demand for safe abortion services[52].

Unsafe abortion was higher in women with the lowest wealth status or women having occupation in the agricultural sector. For those groups, financial barriers could be a factor in the choice of unsafe abortion practices. The Government of Nepal has already made all abortion services freely available since 2017[20], but this might not be enough as users must pay for the cost of medicines. Just making services free may not address all the financial barriers for socio-economically disadvantaged women, and abortion-related direct and indirect costs also need to be addressed while implementing abortion services. Given the findings that women using unsafe abortion practices to end unwanted pregnancy or space or delay childbearing, strengthening family planning service to the wider community is another vital strategy that may help to reduce unsafe abortion practices in Nepal.

## Conclusion

Several factors contribution to unsafe abortion in Nepal. Availability of safe abortion services by establishing safe abortion sites could reduce unsafe abortion practices. Reduction of unintended pregnancy by use of family planning commodities may help women not to choose unsafe abortion practices as a method of child space or delay childbearing. Programmatic efforts should be focussed on access to abortion services to the Mountainous Region, among poor, Dalit and Madhesi and Muslim communities.

## Acknowledgments

The authors are very grateful to Rachael Brennan for the editing support for this manuscript.

## Author Contributions

**Conceptualization:** Resham Bahadur Khatri, Samikshya Poudel, Pramesh Raj Ghimire.

**Data curation:** Resham Bahadur Khatri, Samikshya Poudel, Pramesh Raj Ghimire.

**Formal analysis:** Resham Bahadur Khatri, Pramesh Raj Ghimire.

**Methodology:** Resham Bahadur Khatri, Samikshya Poudel, Pramesh Raj Ghimire.

**Project administration:** Resham Bahadur Khatri, Samikshya Poudel, Pramesh Raj Ghimire.

**Resources:** Resham Bahadur Khatri, Pramesh Raj Ghimire.

**Software:** Resham Bahadur Khatri, Pramesh Raj Ghimire.

**Supervision:** Resham Bahadur Khatri, Pramesh Raj Ghimire.

**Validation:** Resham Bahadur Khatri, Pramesh Raj Ghimire.

**Visualization:** Resham Bahadur Khatri.

**Writing – original draft:** Resham Bahadur Khatri, Samikshya Poudel, Pramesh Raj Ghimire.

**Writing – review & editing:** Resham Bahadur Khatri, Pramesh Raj Ghimire.

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
