## [Decision Letter · Decision Letter 0]

18 Jul 2019

PONE-D-19-14863

Factors associated with unsafe induced abortion practices in Nepal: Pooled Analysis of the 2011 and 2016 Nepal Demographic and Health Surveys.

PLOS ONE

Dear Dr. Khatri,

Thank you for submitting your manuscript to PLOS ONE. After careful consideration, we feel that it has merit but does not fully meet PLOS ONE’s publication criteria as it currently stands. Therefore, we invite you to submit a revised version of the manuscript that addresses the points raised during the review process.

We would appreciate receiving your revised manuscript by 18 August 2019. To enhance the reproducibility of your results, we recommend that if applicable you deposit your laboratory protocols in protocols.io, where a protocol can be assigned its own identifier (DOI) such that it can be cited independently in the future. For instructions see: http://journals.plos.org/plosone/s/submission-guidelines#loc-laboratory-protocols

We look forward to receiving your revised manuscript.

Kind regards,

Russell Kabir, PhD

Academic Editor

PLOS ONE

Journal Requirements:

Reviewers' comments:

Reviewer's Responses to Questions

**Comments to the Author**

1. Is the manuscript technically sound, and do the data support the conclusions?

Reviewer #1: Yes

Reviewer #2: Partly

Reviewer #3: Yes

Reviewer #4: Partly

Reviewer #5: Partly

2. Has the statistical analysis been performed appropriately and rigorously? 

Reviewer #1: Yes

Reviewer #2: No

Reviewer #3: Yes

Reviewer #4: Yes

Reviewer #5: Yes

3. Have the authors made all data underlying the findings in their manuscript fully available?

Reviewer #1: Yes

Reviewer #2: No

Reviewer #3: Yes

Reviewer #4: Yes

Reviewer #5: No

4. Is the manuscript presented in an intelligible fashion and written in standard English?

Reviewer #1: Yes

Reviewer #2: No

Reviewer #3: Yes

Reviewer #4: No

Reviewer #5: Yes

5. Review Comments to the Author

Reviewer #1: Good Effort

Please avoid the word "we or our". You can replace with "This research".

Please add reference number of THREE approval you obtained from Nepal Health Research Council, Nepal; and ICF Marco International Maryland, USA; and DHS program (USA).

Regards

Reviewer #2: Introduction

The paper aimed to (1) provide a national estimate on unsafe abortion rate in Nepal and to (2) examine the factors associated with unsafe induced abortion. Data derived from 2011 and 2016 Nepal Demographic and Health Surveys were analyzed using logistic regression. The authors reported that “women living in mountain Region, urban, poor households; disadvantaged ethnicities such as Dalit or non-Dalit Terai caste, and Muslim, involved in the agricultural sector had higher odds of having unsafe induced abortions compared to their reference’s groups. Women who did not know the location of safe abortion sites had higher odds of unsafe induced abortion, regardless of their knowledge of the legal conditions of abortion,” that “women who lacked knowledge of both place and legal conditions had higher odds of unsafe induced abortions compared to those who knew both,” and that “women who wanted to delay or space childbirth or unwanted birth were associated with higher odds of unsafe induced abortion.”

Merits

While this reviewer has specific comments with regards to some sections, the research on which this article is based is of importance for developing new abortion policies, and once the article is revised, some interesting findings may be gleaned from the data.

Remarks

However, there are a number of general and specific issues that require attention.

Firstly, although the manuscript is generally clear, it would benefit from rephrasing and remolding for clarity and style.

Secondly, the results in the abstract must be reported following specific guidelines established by the international scientific community. For instance, there is no mention of OR, 95% CI, and p values in the abstract.

Further, the authors have placed emphasis on the following factors: place of residence/region, women's age, education (women and their husbands), literacy status, ethnicity, gender (sex of the last child), total number of living male or female children, wealth status, occupation, exposure to mass media on public health issues, knowledge of legal conditions, knowledge of safe abortion place, unmet need for family planning, unintended pregnancy, women's reasons for abortions, and gestational age at abortion.

That is too many variables, some of which may be unrelated to the dependent variable. Though the manuscript attempts to address each of the above-mentioned factors, it failed to take into account the downsides of having models with many independent variables to select from. It is well-known that each irrelevant variable included in the model(s) will decrease the precision of the estimated parameters.

Given the high number of potential predictor variables, it would have been better if the authors had selected the forward stepwise regression (instead of the backward elimination technique used in this paper). The forward stepwise regression is recommended when having a large set of potentially relevant predictor variables. It generates a good sequence of models by allowing to fine-tune them to obtain important information about the quality of the potential predictors.

The backward elimination technique used by the authors is usually applied when there is a modest number of potential predictors, which was not the case here.

Furthermore, the limitation section lacks to mention the limitations of applying the backward elimination method in the selection of potentially relevant variables included in the regression models. Additionally, it is unclear whether the authors used cross-validation to detect potential cases of overfitting and collinearity.

Apart from that, some of the results need to be presented in a different manner, and it is recommend to add more figures/graphs.

Finally, the authors mention in the discussion section that “Among poor, and disadvantaged ethnic communities in Nepal, the contraceptive prevalence rate is high.” They then go on and state that “socioeconomically and disadvantaged ethnic groups have lower contraceptive prevalence rates.” The paper needs coherence.

I hope this review will be helpful and wish the authors the best of luck with their research!

Reviewer #3: This manuscript addresses a relevant topic, such as the determinants of unsafe abortions. It is easy to read and well written. My only concern is about the pooled analysis. Although it is probably necessary in order to obtain a sample big enough, it seems that the main number of unsafe abortions belong to year 2007. As one of the objectives of the study is to propose policies in order to reduce unsafe abortions, conclusions obtained could correspond to the profile of unsafe abortions in 2007, more than in the present. So, I suggest to repeat the analyses conducted in table 2 also in a separate way for each of the years analysed, in order to explore if there are any differences for this period.

Reviewer #4: Congratulations on your work to generate evidence on unsafe abortion practices in Nepal. This cross-sectional study aimed to examine the factors associated with unsafe induced abortion practices in Nepal using 2011 and 2016 Nepal Demographic and Health Surveys. The findings of the study may be useful for policy makers, however, I have some concerns regarding the statistical analysis and discussion of results. In addition, the manuscript needs to be reviewed by a professional English native editor. Some sentences are incomplete or unclear. Please find the detailed comments below by each section.

Introduction:

Overall, introduction needs to be revised. The authors tried to provide data on unsafe abortion at global and national level, however, the authors could present more in greater depth regarding what current evidence is (what do we know now), what is the gap and how this study will fill this gap. The authors need to conduct a proper literature review to provide up to date studies on this topic. The authors could indicate global perspective and findings of other previous studies investigating factors associated with unsafe induced abortions. Later, the authors could mention relevant studies conducted in Nepal and the gaps needed to be addressed. The authors said that there is no study conducted at the national level, however, the authors could mention relevant studies conducted at communication level in Nepal to provide a summary of findings from previous studies.

The authors mentioned that ‘Some studies reported that unsafe abortion rate was higher among women with lower income, ethnic minorities, and lower education’. This looks similar as the finding of this study. Please clearly mention what are the added value of this study.

There is no justification why the authors used 2011 and 2016 NDHS. Please mention why the authors did not use 2001 DHS or 2014 MICS (Multiple Indicator Cluster Surveys).

What is the reference of the sentence ‘The WHO suggests that empirical research on unsafe abortion would help to re-evaluate existing programs as well as formulate appropriate strategies to improve safe abortion services.’? In addition, this sentence does not strengthen the justification of this survey because it is not an empirical research.

Methods:

The authors should provide more details in statistical analysis.

Data source and sample:

1. I suggest the authors to describe DHS in general.

2. what is the total sample size in the end? How did you come to this final sample size?

3. How did you handle missing data?

Independent variables:

For ethnicity, the authors merged some ethnic groups with small sample size into other ethnic groups and said that these groups were similar each other. However, the authors did not provide any evidence with reference on this. Moreover, the authors need to indicate the number of sample size of certain ethnic groups instead of saying ‘small size’. How about Newari origins? This is also one of the unique and major ethnic groups in Nepal.

‘husband education’ is mentioned twice in the Fig 1. conceptual framework. Please remove one.

Statistical analysis:

1. The authors conducted a four staged multivariate logistic regression model but they did not explain why this method is the best to achieve the goal of the study

2. The authors mentioned unadjusted odds ratios as (aOR). Did you mean adjusted odds ratios?

3. The authors did not mention how to choose reference groups when performing logistic regression analysis. Please explain.

4. There are too many independent variables and some are highly interrelated such as women’s education and literacy and women’s occupation and working status. Have the authors checked multicollinearity?

Results:

Descriptive characteristics of the study population:

1. Table 1:

a. It is not clear to me what chai-square means here- is it for categories under unsafe abortion? Please specify.

b. I suggest the authors to revise the table 1. Column percentage and row percentage are mixed so it is confusing.

c. The authors used the symbol, “@ and *”. Need to check whether this is in line with the PLOS ONE guideline.

Unsafe abortion practices in Nepal:

1. The authors mentioned that ‘Over the study period (2007-2016) in Nepal, the total and unsafe abortion rates were 36 (95% CI: 33, 38) and seven (95% CI: 6, 8) per 1000 women aged 15-49 years respectively’. However, there is no table or figure with this data. The authors should present the results with tables or figures. If not in the main manuscript, the authors could provide data in supplementing document.

2. Saying ‘study period 2007-2016’ is confusing. Suggest revising as ‘Data from 2011-2016 NDHS’.

3. The authors mentioned methods of unsafe induced abortions (medical, surgical, etc.), however, there is no data in the table. The authors should present all mentioned data in the table or figure.

Factors associated with unsafe abortion in Nepal:

1. Table 2

a. Why there are three empty rows under predisposing factors?

b. It is not easy to understand the table 2. The authors can consider presenting the results with figure to have a better visualization of results.

2. Even though the authors indicated that they conducted a four staged multivariate logistic regression model, there is no results of model 1-4. What are the results?

3. Also, there is no results regarding this sentence ‘To avoid any statistical bias, the results from the staged model were also checked by: (1) entering only potential risk factors with p-value < 0.20 obtained in the univariate analysis for backward elimination process, and (2) testing the backward elimination method by including all potential risk factors’. It is not clearly mentioned.

Discussion:

In general, the authors should provide a greater explanation of the findings in the discussion section. For instance, it is not clear the implication of the sentence ‘In Mexico, the legal status of abortion varied by state; Mexico City offers abortion up to 13 weeks gestation, whereas in Brazil abortion is legal if pregnancies result from rape or incest or if the life of the pregnant woman or fetus is at risk’. Why is it important and what needs to be done to improve the situation?

Implication:

It would be good to provide relevant references regarding the arguments of the authors. For instance, is there any studies supporting the sentence ‘From the demand side perspective, the community needs to be informed and sensitised about the use of safe abortion services. Moreover, the integration of awareness raising interventions in existing health programs could increase the demand for safe abortion services.’? It may help make the argument strong.

Reviewer #5: Abstract: Can be made more concise

Methods: How was the wealth index calculated? What was the assessment tool used by the NDHS?

How was the rates calculated for abortion rates and unsafe abortion rates?

How was it ensured that the health practitioners who did the abortion were certified for it?

Results: Redesign the table 1 and 2. Make it more clear.

Check the numbers, there are discrepancies. If it is a case of missing data, justify

Limitation and biases has to me mentioned

Discussion needs some more papers which could be more contextual in the countries setting.

6. PLOS authors have the option to publish the peer review history of their article (what does this mean?). If published, this will include your full peer review and any attached files.

Reviewer #1: Yes: Dr Mainul Haque

Reviewer #2: No

Reviewer #3: Yes: Isabel Aguilar

Reviewer #4: No

Reviewer #5: No

---

## [Author Response · Author response to Decision Letter 0]

6 Aug 2019

Point by point responses to the reviewers’ comments

Reviewer #1: Good Effort

Thank you so much for appreciating our work. 

Comment: Please avoid the word "we or our". You can replace with "This research".

Please add reference number of THREE approval you obtained from Nepal Health Research Council, Nepal; and ICF Marco International Maryland, USA; and DHS program (USA).

Response: Agreed, and changed in the revised manuscript as suggested. Regarding approval, we used publicly available secondary data obtained from the DHS program (https://dhsprogram.com/data/available-datasets.cfm). The first author sought approval from MEASURE DHS by online application form to use the data for this study. The details for the application process can be found in the link below: https://dhsprogram.com/data/using-datasets-for-analysis.cfm

Reviewer #2: While this reviewer has specific comments with regards to some sections, the research on which this article is based is of importance for developing new abortion policies, and once the article is revised, some interesting findings may be gleaned from the data.

Thank you for praising our manuscript; and we are pleased to address each of the reviewer’s comments as listed below.

However, there are a number of general and specific issues that require attention.

Thank you for pinpointing important general and specific issues which we have tried to address our best to satisfy the reviewer’s concern.

Comment: Firstly, although the manuscript is generally clear, it would benefit from rephrasing and remolding for clarity and style.

Response: Rephrasing and remodeling have been offered as required. 

Comment: Secondly, the results in the abstract must be reported following specific guidelines established by the international scientific community. For instance, there is no mention of OR, 95% CI, and p values in the abstract.

Response: Thanks. Corrected as suggested (Please see the results section of abstract of the revised manuscript). 

Comment: Further, the authors have placed emphasis on the following factors: place of residence/region, women’s age, education (women and their husbands), literacy status, ethnicity, gender (sex of the last child), total number of living male or female children, wealth status, occupation, exposure to mass media on public health issues, knowledge of legal conditions, knowledge of safe abortion place, unmet need for family planning, unintended pregnancy, women’s reasons for abortions, and gestational age at abortion.

That is too many variables, some of which may be unrelated to the dependent variable. Though the manuscript attempts to address each of the above-mentioned factors, it failed to take into account the downsides of having models with many independent variables to select from. It is well-known that each irrelevant variable included in the model(s) will decrease the precision of the estimated parameters.

Given the high number of potential predictor variables, it would have been better if the authors had selected the forward stepwise regression (instead of the backward elimination technique used in this paper). The forward stepwise regression is recommended when having a large set of potentially relevant predictor variables. It generates a good sequence of models by allowing to fine-tune them to obtain important information about the quality of the potential predictors.

Response: Thank you for the comment. The variables included in this study are important socio-demographic, and maternal factors that are widely used in public health literature as potential predictor variables [1-3]. Given due importance, information on these variables are found in NDHS maternal data file; hence, why included to examine any possible association with the outcome variable. 

The backward elimination technique used by the authors is usually applied when there is a modest number of potential predictors, which was not the case here.

Furthermore, the limitation section lacks to mention the limitations of applying the backward elimination method in the selection of potentially relevant variables included in the regression models. Additionally, it is unclear whether the authors used cross-validation to detect potential cases of overfitting and collinearity.

Response: We consulted this with a mathematical and applied statistician. We kindly disagree with the reviewer, and we think the reviewer meant the contrary because ‘Overfitting’ occurs when a model is having too many parameters (variables). The staged model was introduced in this study to avoid the issue of overfitting. Also, we also tested our stage modelling approach by using both forward and backward elimination method; the three methods found the same variables to be significantly associated with unsafe abortion. We have also tested and reported multi-collinearity (please see last paragraph of page 8, statistical analysis section. Our approach of statistical analysis is consistent with previous studies [4-6]. 

Apart from that, some of the results need to be presented in a different manner, and it is recommend to add more figures/graphs.

Response: Our apology. We would be grateful if the reviewer can be more specific on his/her comment that which results are recommended to be in the figure. However, the way we have presented our results are easy for readers to navigate. In addition, this style of presenting the results are widely found in recent public health literature. 

Finally, the authors mention in the discussion section that “Among poor, and disadvantaged ethnic communities in Nepal, the contraceptive prevalence rate is high.” They then go on and state that “socioeconomically and disadvantaged ethnic groups have lower contraceptive prevalence rates.” The paper needs coherence.

Response: Thank you for this mistake, we have corrected this.

I hope this review will be helpful and wish the authors the best of luck with their research!

Response: We are grateful to the reviewer; and we have addressed almost all the comments from reviewer 2 while taking other reviewers comments into account. Thank you very much for wishes.

Reviewer #3: This manuscript addresses a relevant topic, such as the determinants of unsafe abortions. It is easy to read and well written. My only concern is about the pooled analysis. Although it is probably necessary in order to obtain a sample big enough, it seems that the main number of unsafe abortions belong to the year 2007. As one of the objectives of the study is to propose policies in order to reduce unsafe abortions, conclusions obtained could correspond to the profile of unsafe abortions in 2007, more than in the present. So, I suggest repeating the analyses conducted in table 2 also in a separate way for each of the years analysed, in order to explore if there are any differences for this period.

Response: We agree with the reviewer, and the aim of using pooled datasets was to increase the sample size so as to increase the statistical power to help detecting any statistical differences in the course of statistical modelling; consistent with previous studies [4-6]. As per the reviewer’s suggestion, we have however accommodated this as a limitation of the study (Please see in the revised manuscript which reads as: ‘Due to the small sample size, this study could not do a separate analysis for each of the survey wave (NDHS 2016 and NDHS 2011) for absolute comparison’). 

Reviewer #4: Comment: Congratulations on your work to generate evidence on unsafe abortion practices in Nepal. This cross-sectional study aimed to examine the factors associated with unsafe induced abortion practices in Nepal using 2011 and 2016 Nepal Demographic and Health Surveys. The findings of the study may be useful for policy makers, however, I have some concerns regarding the statistical analysis and discussion of results. In addition, the manuscript needs to be reviewed by a professional English native editor. Some sentences are incomplete or unclear. Please find the detailed comments below by each section.

Response: the manuscript has been reviewed by a professional editor. Language has been edited as suggested. 

Introduction: 

Comment: Overall, introduction needs to be revised. The authors tried to provide data on unsafe abortion at global and national level, however, the authors could present more in greater depth regarding what current evidence is (what do we know now), what is the gap and how this study will fill this gap. The authors need to conduct a proper literature review to provide up to date studies on this topic. The authors could indicate global perspective and findings of other previous studies investigating factors associated with unsafe induced abortions. 

Response: Thank you for the feedback. We reviewed relevant literature of global context and have written on page 3, first and second paragraphs).

Later, the authors could mention relevant studies conducted in Nepal and the gaps needed to be addressed. The authors said that there is no study conducted at the national level, however, the authors could mention relevant studies conducted at communication level in Nepal to provide a summary of findings from previous studies.

Response: Thank you for the feedback. We reviewed relevant literature of Nepalese context and have written in full paragraph (page 4, first and second paragraphs):

Comment: The authors mentioned that ‘Some studies reported that unsafe abortion rate was higher among women with lower income, ethnic minorities, and lower education’. This looks similar to the finding of this study. Please clearly mention what are the added value of this study.

Response: Thank you for the comment, and this has been addressed in the revised manuscript (Please see last paragraph of page 4).It has been corrected.

Comment: There is no justification why the authors used 2011 and 2016 NDHS. Please mention why the authors did not use 2001 DHS or 2014 MICS (Multiple Indicator Cluster Surveys).

Response: We have not included previous surveys (NDHS 2001 and NDHS 2006) because those surveys lacked information on abortion services. In facts, those surveys have not included questions on abortion services. 

Comment: What is the reference of the sentence ‘The WHO suggests that empirical research on unsafe abortion would help to re-evaluate existing programs as well as formulate appropriate strategies to improve safe abortion services.’? In addition, this sentence does not strengthen the justification of this survey because it is not an empirical research.

Responses: Thank you for the comment. References are provided for the arguments suggested. The word empirical research in this manuscript was used to reflect the practical research; and the findings from nationally representative NDHS can be the useful instrument to inform policy and practice. 

Methods: 

The authors should provide more details in statistical analysis.

Data source and sample: 

1. I suggest the authors to describe DHS in general.

Responses: description of DHS is provided in page 5 second paragraph.

2. What is the total sample size in the end? How did you come to this final sample size?

Response: total sample size was 911. These are pooled data of NDHS 2011 AND 2016. Detailed descriptions are provided on page 5, last two paragraphs.

3. How did you handle missing data?

Response: A total of 45 missing values were excluded from the multivariate logistic regression analysis, and this has been stated in the methods section of the revised manuscript (please see line … of page …..). In addition, we have mentioned in the limitation that we could not include gestational period, an important confounder, into the adjusted regression model because of huge missing values (20%) which in case of inclusion could bias the result (please see line … of the study limitation section of the revised manuscript). 

Independent variables: 

For ethnicity, the authors merged some ethnic groups with small sample size into other ethnic groups and said that these groups were similar each other. However, the authors did not provide any evidence with reference on this. Moreover, the authors need to indicate the number of sample size of certain ethnic groups instead of saying ‘small size’. How about Newari origins? This is also one of the unique and major ethnic groups in Nepal.

Response: References are provided as suggested for ethnic categorization. Like other studies[4, 6], Newari ethnic group also included into Janajati ethnic group.

‘Husband education’ is mentioned twice in the Fig 1. Conceptual framework. Please remove one.

Response: This has been corrected.

Statistical analysis: 

1. the authors conducted a four staged multivariate logistic regression model but they did not explain why this method is the best to achieve the goal of the study.

Response: The four-stage technique was adopted based on four-level of data that can be divided based on its proximity to the outcome [3, 7]. This has been addressed in the revised manuscript (Please last paragraph of page 8, statistical analysis subheading under methods section). This approach is also consistent with previous studies that used Nepal DHS data [3-6] .

2. The authors mentioned unadjusted odds ratios as (aOR). Did you mean adjusted odds ratios?

Responses: Yes, aOR means adjusted odds ratio. It has been corrected.

3. The authors did not mention how to choose reference groups when performing logistic regression analysis. Please explain.

Response: references group are chosen considering the possibility of a better interpretation of the findings. In the most of cases, we choose the advantaged category as reference groups.

4. There are too many independent variables and some are highly interrelated such as women’s education and literacy and women’s occupation and working status. Have the authors checked multicollinearity?

Responses: We checked multicollinearity using Variation Inflation Factor test; however, there was not find any multi-collinearity of the variables.

Results: 

Descriptive characteristics of the study population: 

1. Table 1: 

a. It is not clear to me what chai-square means here- is it for categories under unsafe abortion? Please specify.

Responses: It is chi-square p-value obtained from cross-tabulation of each independent variables and unsafe abortion. It has been corrected in the table.

b. I suggest the authors to revise the table 1. Column percentage and row percentage are mixed so it is confusing.

Response: It has been corrected; column percentage is deleted. Now each row per cent indicates the % of unsafe abortion out of total abortion in that category. 

c. The authors used the symbol, “@ and *”. Need to check whether this is in line with the PLOS ONE guideline.

Response: PLOS ONE Guideline allows those symbols; however, we have deleted in the revised manuscript.

Unsafe abortion practices in Nepal: 

1. the authors mentioned that ‘Over the study period (2007-2016) in Nepal, the total and unsafe abortion rates were 36 (95% CI: 33, 38) and seven (95% CI: 6, 8) per 1000 women aged 15-49 years respectively’. However, there is no table or figure with this data. The authors should present the results with tables or figures. If not in the main manuscript, the authors could provide data in supplementing document.

Response: We have revised methods section how it was abortion rates were calculated (see page 8 under statistical subheading). Simply abortion rates are the total numbers of abortions per thousand women of reproductive age (15-49 years). It is calculated using formula total numbers of abortion (or unsafe abortions for unsafe abortion rate) divided by total numbers of women interviewed and multiplied by 1000. .

Best is give him the table as supplementary as discussed previously. 

2. Saying ‘study period 2007-2016’ is confusing. Suggest revising as ‘Data from 2011-2016 NDHS’.

Response: It has been revised as suggested.

3. The authors mentioned methods of unsafe induced abortions (medical, surgical, etc.), however, there is no data in the table. The authors should present all mentioned data in the table or figure.

Response: It has been corrected as suggested. This study aimed to identify factors associated with unsafe abortion, and putting this information in the table does not suit this study. Therefore we used this in the textual form for a general overview. 

Factors associated with unsafe abortion in Nepal: 

1. Table 2

a. Why there are three empty rows under predisposing factors?

Response: it has been corrected.

b. It is not easy to understand the table 2. The authors can consider presenting the results with figure to have a better visualization of results.

Response: We think the way we have presented the results in the table is good for lay health workers as well as general readers. Additionally, p values of adjusted odds ratio were made bold which indicate significance in the final model at p<0.05.

2. Even though the authors indicated that they conducted a four staged multivariate logistic regression model, there is no results of model 1-4. What are the results?

Response: Putting results from 4 stages in the paper looked very busy. We decided to use the final model for readers to navigate easily; and this has been done previously [8]. 

3. Also, there is no results regarding this sentence ‘To avoid any statistical bias, the results from the staged model were also checked by: (1) entering only potential risk factors with p-value < 0.20 obtained in the univariate analysis for backward elimination process, and (2) testing the backward elimination method by including all potential risk factors’. It is not clearly mentioned.

Response: We employed alternative regressions technique to confirm/validate the estimates but we found staged regression technique have provided precise estimates than other techniques. 

Discussion: 

In general, the authors should provide a greater explanation of the findings in the discussion section. For instance, it is not clear the implication of the sentence ‘In Mexico, the legal status of abortion varied by state; Mexico City offers abortion up to 13 weeks gestation, whereas in Brazil abortion is legal if pregnancies result from rape or incest or if the life of the pregnant woman or fetus is at risk’. Why is it important and what needs to be done to improve the situation?

Response: Thank you for your suggestions. We provided those statement to support our findings of socioeconomically poor women have higher odds of unsafe abortion. Poor Mexican and Brazilian women have also had a higher unsafe abortion because of legal barriers in those countries.

Implication: 

It would be good to provide relevant references regarding the arguments of the authors. For instance, is there any studies supporting the sentence ‘From the demand side perspective, the community needs to be informed and sensitized about the use of safe abortion services. Moreover, the integration of awareness raising interventions in existing health programs could increase the demand for safe abortion services.’? It may help make the argument strong.

Response: many thank you for your suggestions. We have provided relevant references on our important arguments, including the above statements. 

Reviewer #5: Abstract: Can be made more concise

Response: it has been revised as suggested.

Methods: How was the wealth index calculated? What was the assessment tool used by the NDHS?

Response: In NDHS, wealth quintiles were calculated using principal component analysis of 40 households’ asset items. In this research, household wealth quintile were categorised into three groups: the bottom 40% was referred to as poor households, the next 40% as the middle households and the top 20% as rich households.

How was the rates calculated for abortion rates and unsafe abortion rates?

Response: We have revised methods section how it was abortion rates were calculated (last paragraph of page 8, statistical analysis subheading). Simply abortion rates are the total numbers of abortions per thousand women of reproductive age (15-49 years). It is calculated using formula total numbers of abortion (or unsafe abortions for unsafe abortion rate) divided by total numbers of women interviewed and multiplied by 1000.

How was it ensured that the health practitioners who did the abortion were certified for it?

Response: In Nepal, nurses and doctors get training for abortion services, so we have included doctors and nurse as skilled providers assuming there were certified providers. 

Results: Redesign the table 1 and 2. Make it clearer.

Response: tables have been revised and made more readable.

Check the numbers, there are discrepancies. If it is a case of missing data, justify

Response: It has been checked and corrected if needed.

Limitation and biases have to be mentioned

Response: it has been revised as suggested. A full paragraph has been developed for limitations and strengths.

Discussion needs some more papers which could be more contextual in the countries setting.

 Responses: it has been revised as suggested. Many thank you for your insightful comments. 

1. Mohan, D., et al., Analysis of dropout across the continuum of maternal health care in Tanzania: findings from a cross-sectional household survey. 2017. 32(6): p. 791-799.

2. Joshi, C., et al., Factors associated with the use and quality of antenatal care in Nepal: a population-based study using the demographic and health survey data. 2014. 14(1): p. 94.

3. Khanal, V., et al., Under-utilization of antenatal care services in Timor-Leste: results from Demographic and Health Survey 2009–2010. 2015. 15(1): p. 211.

4. Poudel, S., et al., Trends and factors associated with pregnancies among adolescent women in Nepal: Pooled analysis of Nepal Demographic and Health Surveys (2006, 2011 and 2016). 2018. 13(8): p. e0202107.

5. Akombi, B.J., et al., Child malnutrition in sub-Saharan Africa: A meta-analysis of demographic and health surveys (2006-2016). 2017. 12(5): p. e0177338.

6. Ghimire, P.R., et al., Factors associated with perinatal mortality in Nepal: evidence from Nepal demographic and health survey 2001–2016. 2019. 19(1): p. 88.

7. Victora, C.G., et al., The role of conceptual frameworks in epidemiological analysis: a hierarchical approach. 1997. 26(1): p. 224-227.

8. Ghimire, P.R., et al., Socio-economic predictors of stillbirths in Nepal (2001-2011). PloS one, 2017. 12(7): p. e0181332.

---

## [Editor Report · Decision Letter 1]

20 Sep 2019

Factors Associated with Unsafe Abortion Practices in Nepal: Pooled Analysis of the 2011 and 2016 Nepal Demographic and Health Surveys

PONE-D-19-14863R1

Dear Ms. Khatri,

We are pleased to inform you that your manuscript has been judged scientifically suitable for publication and will be formally accepted for publication once it complies with all outstanding technical requirements.

With kind regards,

Russell Kabir, PhD

Academic Editor

PLOS ONE
---

## [Editor Report · Acceptance letter]

30 Sep 2019

PONE-D-19-14863R1 

Factors Associated with Unsafe Abortion Practices in Nepal: Pooled Analysis of the 2011 and 2016 Nepal Demographic and Health Surveys 

Dear Dr. Khatri:

I am pleased to inform you that your manuscript has been deemed suitable for publication in PLOS ONE. Congratulations! Your manuscript is now with our production department. 

With kind regards,

on behalf of

Dr. Russell Kabir 

Academic Editor

PLOS ONE